# EDUROVs: A Low Cost and Sustainable Remotely Operated Vehicles Educational Program

Xavier Cufí [1,*], Albert Figueras [1], Eduard Muntaner [1], Remei Calm [1], Eduardo Quevedo [2], Daura Vega [3], Josefina Loustau [4], José Juan Gil [4] and Joaquín H. Brito [4]

1   Institute of Computer Vision and Robotics, University of Girona, E.P.S., 17071 Girona, Spain; albert.figueras@udg.edu (A.F.); eduard.muntaner@udg.edu (E.M.); remei.calm@udg.edu (R.C.)
2   Institute for Applied Microelectronics (IUMA), University of Las Palmas de Gran Canaria (ULPGC), 35017 Las Palmas de Gran Canaria, Spain; equevedo@iuma.ulpgc.es
3   Chemistry Department, University of Las Palmas de Gran Canaria (ULPGC), 35017 Las Palmas de Gran Canaria, Spain; daura.vega@ulpgc.es
4   Oceanic Platform of the Canary Islands (PLOCAN), 35214 Telde, Spain; josefina.loustau@plocan.eu (J.L.); jose.gil@plocan.eu (J.J.G.); joaquin.brito@plocan.eu (J.H.B.)
*   Correspondence: xavier.cufi@udg.edu; Tel.: +34-972-41-8757

**Abstract:** EDUROV is an educational underwater robot proposal from the researchers of the Oceanic Platform of Canary Islands (PLOCAN) and the Computer Vision and Robotics research group of the University of Girona (VICOROB), launched in January 2012 with the support of the Spanish Foundation for Science and Technology (FECyT). This program has evolved in the last decade in order to make it more sustainable, allowing the teleoperation of underwater vehicles from anywhere in the world. EDUROVs have passed through several phases, beginning with a basic electronics robot, followed by the incorporation of open-source electronic prototyping platforms and finally reaching the current state of teleoperation. Results based on 1–5 Likert scale questions show that both students and teachers consider the program useful to introduce technical and scientific concepts. It is concluded that the use of low-cost materials and tools that are easy to obtain, following education on sustainability approaches, also makes them possible for use in high schools, and science teachers can easily carry out the activity. Moreover, the possibility of remote teleoperation of underwater vehicles, together with the collaboration among groups of students in different locations that are in contact through these online tools, allows one to motivate students to work on the project from a different perspective.

**Keywords:** underwater robotics educational project; workshops for students; attracting young people to technology and science; exploring underwater environments

## 1. Introduction

Oceans, seas and coastal areas form an integrated and essential component of the Earth's ecosystem and are critical to sustainable development. They cover more than two-thirds of the Earth's surface and contain 97% of the planet's water. Over three billion people depend on marine and coastal resources for their lives. In addition, oceans are also the primary regulator of the global climate, an important sink for greenhouse gases and they provide water and oxygen to the Earth. Finally, oceans host huge reservoirs of biodiversity [1].

Conservation and sustainable use of the oceans, seas and marine resources is the 14th Sustainable Development Goal [2]. One of the targets of this goal, specifically the tenth, is to increase scientific knowledge, develop research capacity and transfer marine technology, in order to improve ocean health and to enhance the contribution of marine biodiversity to the evolution of developing countries.

For all of these reasons, exploring underwater environments, and specifically oceans, is always a very important challenge for humanity. The harsh marine conditions, depth and

extent make it extremely difficult for humans to be able to gain knowledge and develop activities in it. This challenge is very attractive and appealing to young people, and it can be a strong stimulus to foster their vocation to technological and scientific disciplines. On the other hand, Education for Sustainability is an educational approach that aims to develop students, schools and communities with the values and the motivation to take action for sustainability [3]. Education for Sustainability aims to build awareness and knowledge of sustainability issues but also to develop students and schools that are able to think critically, innovate and provide solutions towards more sustainable patterns of living.

Young students are naturally very curious about the world around them. Constructionism is a learning theory that argues that learning is an active process of knowledge construction. This construction of knowledge occurs much more effectively when objects and artifacts that are tangible and shareable are constructed. Unfortunately, as they grow older, they tend to perceive technology and engineering issues as difficult, unknown and strange, which causes them to stop considering careers related to technology, engineering, and math. The activity described in this proposal is designed for young people and can be addressed at the public in general [4,5].

The educational experiences of the authors with young people confirm these ideas without any doubt. It is truly believed that doing hands-on activities is ideal for promoting meaningful learning. In particular, these "constructionist" project-based activities are considered highly relevant for exposing students to the fields of Science, Technology, Engineering, and Mathematics (STEM) [6] and if some kind of artistic manifestation is added (STEAM), these activities are even more powerful [7,8].

The Massachusetts Institute of Technology (MIT) SeaPerch Program was created by the MIT Sea Grant College Program in 2003. Now, SeaPerch program is an initiative from RoboNation [9] with the objective of reducing traditional barriers to participate in robotics programs and promotes opportunities to engage students and educators in inquiry-based learning with real-world applications. As an educational program, SeaPerch introduces students to basic engineering, design, and science concepts [10]. As a fun hands-on project, SeaPerch engages students and fosters key 21st century skills including critical thinking, collaboration, and creativity. Another very interesting and much related initiative is MATE ROV Competition [11]. It is an underwater robotics challenge that engages a lot of learners and volunteers each year, as well. During year 2021, the competition is challenging students to tackle problems that impact the entire world: plastics clogging the rivers, lakes, waterways, and ocean; climate change raising ocean temperatures, affecting the health of coral reefs; and contaminants in the waterways. SeaPearch and MATE programs were a huge inspiration for EDUROV project. EDUROV is a proposal from the researchers of the Oceanic Platform of Canary Islands (PLOCAN) and the Computer Vision and Robotics research group of the University of Girona (VICOROB), launched in January 2012 with the support of the Spanish Foundation for Science and Technology (FECyT).

## 2. Theoretical and Practical Background

The main idea of EDUROV project is to create small-scale, simple but functional prototypes using everyday materials whenever possible. A Remote Operated Vehicle (ROV) is an unmanned submarine vehicle controlled by a command console attached to the vehicle by an umbilical cord. These ROVs are equipped with engines for propulsion and can be also equipped with sensors of different characteristics, underwater cameras and intervention widgets such as mechanical arms.

The core of EDUROV project are the workshops with the students. In that context, they perform different types of practical work and experiments that are very appropriate to validate theoretical explanations of concepts related to Physics, Science, Technology and Engineering. At the same time, and with as much or more importance, they are also working with very important aspects of learning, such as learning to be and act autonomously, learning to think and communicate, learning to discover and take initiative and learning to live and inhabit the world. Creativity, critical thinking and the values

associated to teamwork are key issues. It is a very clear example of project-based learning and "maker" culture, with a profound focus on the do-it-yourself philosophy. The learning carried out in these workshops is embedded in different phases of underwater robotic vehicle design and development. Concepts such as mass distribution on the vehicle, chassis design, location of propellers, aspects of buoyancy, design of electrical circuits, control of DC motors, waterproofing the different elements and adapting them to the environment, are worked on during the development of the underwater vehicles [12].

In addition, procedures that are typical of the development of engineering projects are also put into practice. Aspects such as analysis of documentation, technical discussions, work planning, teamwork, use of different types of tools of different complexity in an effective and safe way, testing of the different subsystems, design of tests and trials to analyze the performance of the vehicle, detection and correction of faults, design of experiments, project documentation, improving communication skills, are also considered.

Also within the framework of the EDUROV initiative and according to the maker philosophy, two projects for the promotion of scientific, technological culture and innovation of the Spanish government (ROVINO and ROVSTEAM) have been developed during the last two years, to reinforce and expand EDUROVs. In these projects, the way to incorporate different kind of technological elements that allow working with students of different educative levels through a project-based learning methodology was analyzed, from primary education to university covering almost all the educational spectrum, even non-standard education. The following improvements were considered in the framework of these projects: programming of microcontroller boards, 3D printing of parts of the underwater vehicle, wireless teleoperation, studying the use of different kind of sensors, design of simple tools for intervention over the environment, etc.

These activities are also used to explain and discuss with the participating students about the different research projects and underwater vehicles developed by the marine community. It is shown that underwater robotics is a key point to develop underwater technological and scientific projects, related to biology, geology, archeology, natural resources and building underwater facilities. This diversity makes the project even more transversal. The interaction and work of students with senior researchers at the university or research centers can also be a very motivating factor in increasing their interest in science and technology.

Two research centers that have these topics in its normal activity, and that are completely complementary under the point of view of the objectives of the EDUROV project, are PLOCAN and VICOROB.

The Oceanic Platform of the Canary Islands (PLOCAN) is a Singular Scientific and Technical Infrastructure formed by a series of specialized facilities that, together, provide access to study or test excellent and innovative scientific-technological concepts and devices in coastal and ocean environments. It includes a Test Bank and a multipurpose ocean platform. These facilities provide data provisioning, operations and accommodation services for experiments or new devices. Its main function is to accelerate research, technological development and innovation in the marine-maritime sector, as well as to provide the critical facilities necessary for the international scientific and industrial community to carry out its experiments.

PLOCAN includes among its objectives to bring to the population the intrinsic values that technology and marine science provides, through the promotion and dissemination of training, specialization and educational capacities.

Furthermore, it includes gaining knowledge on the technologies and facilities that are available for study, research and innovation in the marine and maritime sector, which enhance the possibility of increasing the motivation, participation and commitment of citizens, from an early age [13,14]. Moreover, PLOCAN is supported in the educational robotics initiative by researchers of the University of Las Palmas de Gran Canaria (ULPGC) located on the Gran Canaria island [15,16].

VICOROB is a research group at the University of Girona specialized in computer vision and robotics that combines top level research with technology transfer, focusing on basic and applied research projects [17]. VICOROB has three main research lines represented by three labs. Girona Underwater Vision and Robotics lab that has become a benchmark in Europe for the design and construction of autonomous underwater vehicles, and the development of cutting-edge software for the processing of visual and acoustic underwater images. The Image Analysis and 3D perception lab is committed to developing and optimizing methods for analysis of data, with particular focus on the study of medical images, and mobile robot navigation, and UdiGitalEdu lab main objective is to develop projects in the areas of education and technology with a distinctive multidisciplinary approach. UdiGitalEdu designs activities and workshops that mix technology, science and art with the aim of encouraging creativity, critical thinking and teamwork.

UdiGitalEdu has huge experience in developing workshops related with the development of low-cost underwater vehicles since 2008. More than 50 three-day workshops have been held, involving the participation of more than 700 high school students, and more than 200 robots built. The experience has been published in scientific journals, and in international congresses, and demonstrations have been carried out at international and national events. This initiative obtained the Jaume Vicenç Vives award from the Generalitat de Catalunya to the collective modality in 2018 for its work in disseminating science and promoting technological and engineering vocations among university and high school students [18].

VICOROB and PLOCAN research groups have followed different, but complementary, philosophies for the development of the project, in the respective educative communities. VICOROB proposes holding the workshops entirely at the Underwater Robotics Research Center (CIRS). The students fully develop the underwater robots for 3 days at the Research Center with the direct contact and support of the CIRS researchers participating in the project and the support of the teachers from the secondary schools who accompany the students. PLOCAN proposes sending the material to the different educational centers participating in the project and conducting prior training for the teachers of these centers. These different centers are located in the different islands of the Canary archipelago or in different cities of Spain. An annual final event is held, where all the secondary schools attend, all the underwater vehicles developed are shown, and the results are discussed.

The EDUROV project consists of carrying out a wide variety of activities whose central axis is to support the holding of workshops with students, and to appropriately disseminate the project in educational centers, to the scientific community, and to the public in general:

- Teacher training sessions;
- Visits to secondary schools to help to develop the project;
- Demonstrations in public museums for the general public;
- Participation in international and national technology fairs;
- Meetings and events of secondary schools participating in the project;
- Conferences for the general public;
- Final degree projects;
- High school research works;
- Labor Stays for students in the framework of the project.

## 3. Material and Methods

The EDUROV project has gone through different phases where the underwater vehicle has been incorporated different technological elements that have allowed the contact of the students with a greater number of subjects and fields of knowledge, with the consequent incorporation of students of different educational levels, increasing in this way the potential school audience to which the project is addressed.

In the beginning, the main idea of EDUROV project was to create small-scale, simple but functional prototypes of underwater vehicles using everyday materials. The proposed remote operated vehicles (ROVs) are unmanned submarine vehicles controlled by a com-

mand console based on a joystick and two pushbuttons attached to the vehicle by an umbilical cord, as it is shown in Figure 1 [19,20]. Underwater vehicles are built with a very basic set of considerations in mind. The chassis must be able to house the two horizontal motors and the vertical motor, and it must be able to house in the upper part the floating elements that are required to balance the vehicle underwater. The size is that of a shoe box, approximately. This size is appropriate given the power of the DC motors used. It is considered as the first phase of the EDUROV project.

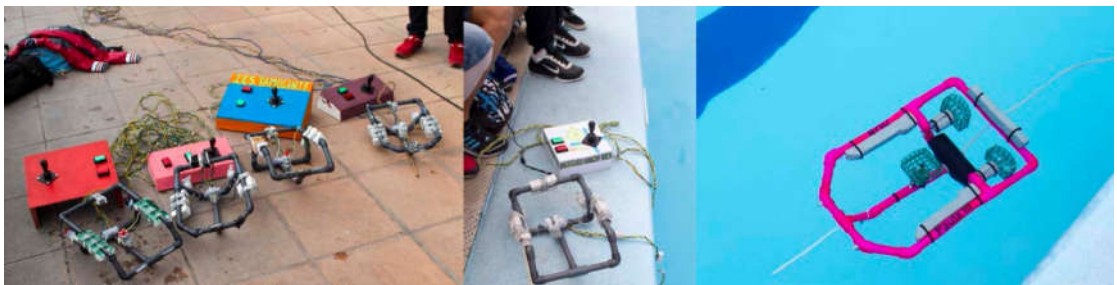

**Figure 1.** Different models of ROVs developed by the students.

In this case, the knowledge and skills required for the development of these underwater vehicles are those corresponding to students in the third or fourth year of Compulsory Secondary Education (ESO, in Spain), which corresponds to students between 14 and 15 years old [8,21].

Figure 2 shows the front covers of the two handbooks produced on this first stage by the team, to help teachers and students to develop the ROVs [19,20].

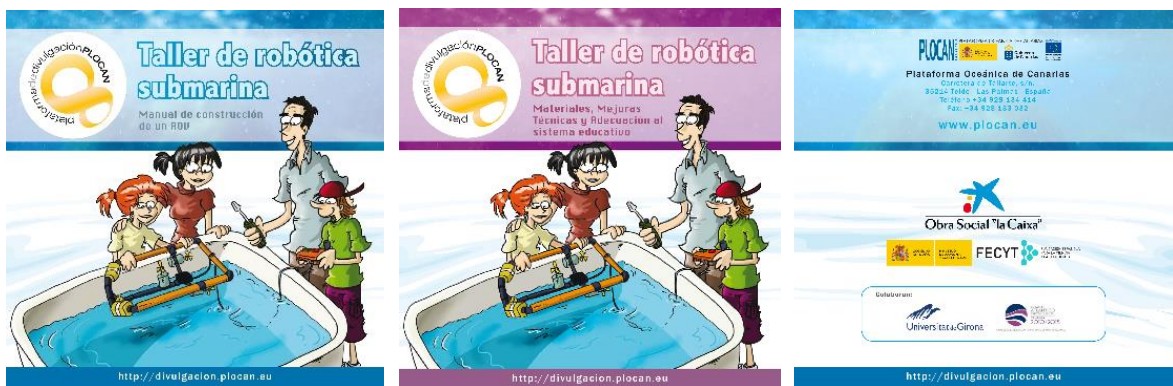

**Figure 2.** Support publications for EDUROV project (first phase).

The second technological phase of the project begins with the incorporation of open-source electronic prototyping platforms, such as Arduino, programmable through free software [22]. They allow the replacement of the original console by a system based on a microcontroller connected to H-bridges to control the 12VDC motors of the underwater robot. These elements allow the evolution of the project in several ways:

- The joystick and buttons are inputs to this microprocessor system that must be controlled by program. If high-level software is used, and these new elements are studied in detail, the project is very appropriate for high school students.
- For students in the last years of primary school and first years of Compulsory Secondary Education (ESO), the set underwater vehicle—microcontroller system could be considered as a black box. In this case, students can get started in the world of programming using visual programming methods such as Scratch to control the movements of the robot [23]. Scratch is a free programming language and online

community where you can create your own interactive stories, games, and animations, and interact with external devices [24].

- A mobile device (tablet or smartphone) that performs the functions of the console through an app that the students may have developed can replace the joystick and buttons. In this case, the robot movement instructions are provided from the mobile device using a Bluetooth link with the microcontroller system [25]. The use of a mobile device and the programming of a simple app is very attractive and suitable for high school students and students in the initial courses of engineering degrees.

In order to help educational centers develop these improvements proposed during this second phase, a new version of the handbook was produced including all the new contents related to the integration of electronic prototyping platforms and the evolution of the project [22].

In this second technological phase of the project, autonomous underwater vehicles-AUV can be also developed. In this case, the movements of the underwater robots are planned and programmed in advance (mission scheduling). There is no operator intervention during the course of the mission. This implies the absence of the console and the umbilical cord, as well as having to ship the batteries that provide the power supply of the system, and the microcontroller system. Shipping these systems on the underwater vehicle requires waterproofing these elements, and this involves a series of changes in the standard structure of the vehicle. Normally, students of engineering degrees develop these autonomous vehicles.

In the third technological phase of the EDUROV project, a microcontroller system has been incorporated that is a benchmark in the development of IoT devices. This system allows the teleoperation of underwater vehicles from anywhere in the world. The microcontroller system is programmed as a Web server, receiving remote requests from computers, smartphones, etc. as it is shown in Figure 3. This technological phase of the project is suitable for high school and engineering students.

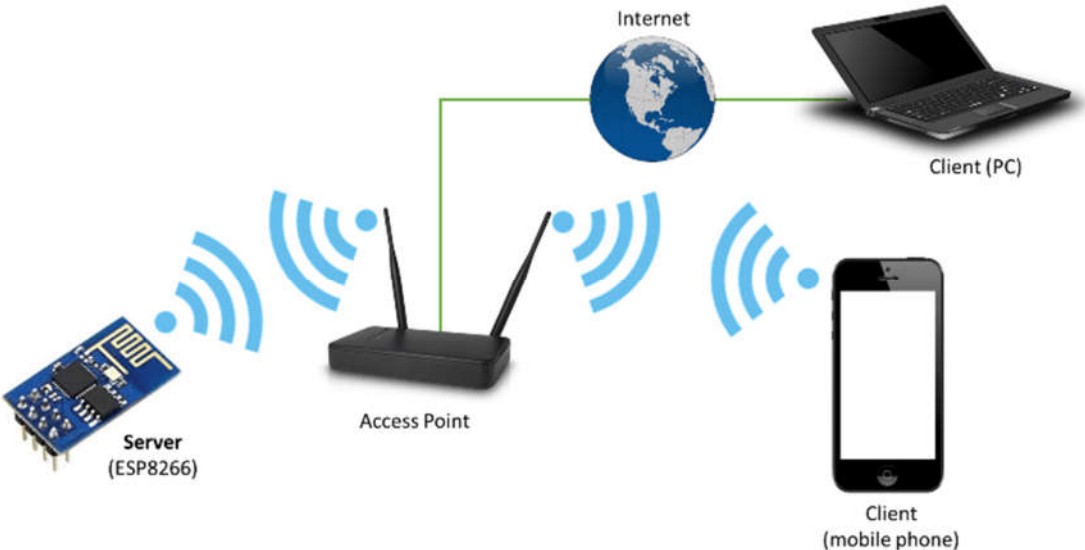

**Figure 3.** ESP8266 microcontroller operating as a Web Server [26].

Currently, the project is being developed simultaneously in all the specified technological phases since the objective is to reach students of any educational level.

As it is shown in the Results section, two surveys are presented in order to consider the perception of students and teachers participating on workshops and program events respectively. With respect the representativeness of the sample, the population considered in this study has been 2 million students, as it is the total amount of secondary school students in the Canary Islands at the time the survey was conducted.

Equation (1) describes the minimum representative sample given a targeted confidence level (which provides a score value ($z$)), margin of error ($\varepsilon$), population proportion ($p$) and population size ($N$). For a confidence value of 95% (1.96 score), a margin of error of 5% and an assumed a population proportion of 50%, the minimum sample needed to be representative would be of 384 students. In case of students, the result can be applied in any case to a higher population, as the considered population is so high that the denominator in the formula can be considered as "1".

$$s = \frac{\frac{z^2 \cdot p(1-p)}{\varepsilon^2}}{1 + \frac{z^2 \cdot p(1-p)}{\varepsilon^2 N}} \tag{1}$$

The survey developed for this study was shared with students, comprising a sample of 384 students. In terms of teachers, it is not clear as to how many teachers conduct these types of activities in the classroom, so a standard value of 205 teachers seems to be enough for a maximum population of 400 teachers.

The surveys were designed in order to evaluate the following aspects:

- Student survey
  - Logistics and Facilities: Questions 1–5;
  - Underwater robotics topics: Questions 6 and 12–14;
  - Activities: Questions 7–9;
  - Teamwork: Questions 10–11;
  - Materials and Tools: Questions 15–16;
  - Teachers: Questions 17–18;
  - General assessment: Questions 19–22.
- Teacher survey:
  - Timing: Questions 1, 7, 9 and 10;
  - Organization: Questions 2–3;
  - Facilities: Questions 4 and 15;
  - Project and contents: Questions 5, 6, 8 and 11;
  - Trainers and Students: Questions 12–14;
  - General assessment: Questions 16–17.

To validate the reliability of these surveys as suitable data collecting instruments, the Cronbach's alpha method was implemented obtaining coefficients higher than 0.65 in all cases. These scores are considered as adequate according to authors such as Huh et al. [27]. These authors consider that the reliability value in exploratory research should be equal or higher than 0.6. Other authors, such as, for instance, Nunnally [28], state that, for early stages of a research, a value of 0.5 or 0.6 would be sufficient. Therefore, the surveys used as instruments in this paper count on a high reliability rate.

## 4. Results

At the end of each workshop, the students fill out an evaluation form that collects different aspects related to the development of the activity and the degree of learning that the students have acquired. In addition, the students' responses are used to make pedagogical and procedural improvements in the different phases of the development of the workshops. Table 1 shows the results obtained from the responses of 384 students from different secondary schools who participated in the experience [12]. Not all students answered the questionnaire. The results obtained are considered very significant.

**Table 1.** Questionnaire for students participating on the workshops and score.

| Questions | Score (1–5) |
|---|---|
| Q1. How easy was to reach the facility by public transportation | 4.11 |
| Q2. How do you rate the scheduling of the activities | 3.33 |
| Q3. Evaluate the facilities: where workshop took place | 4.17 |
| Q4. Evaluate facilities: restaurants, surroundings, free time spaces … | 4.50 |
| Q5. Evaluate resources to do the workshop efficiently/properly | 4.72 |
| Q6. Is it the topic underwater robotics interesting? | 4.70 |
| Q7. Was it easy to build the ROV? | 2.88 |
| Q8. Did you like the introductory talk? | 3.35 |
| Q9. Did you like the activity to create work teams | 3.00 |
| Q10. Do you think that your team was efficient? | 2.94 |
| Q11. Did you enjoy working with your team? | 3.50 |
| Q12. Did you like the talk about underwater robotics | 3.47 |
| Q13. Where the theoretical concepts explained during the workshop adequate for your level of knowledge? | 3.83 |
| Q14. Did you find it easy to understand the basic principles of underwater robotics? | 3.28 |
| Q15. Did you find interesting to work with materials and tools available during the workshop? | 4.11 |
| Q16. Did you feel safe working with materials and tools during the workshop? | 4.33 |
| Q17. Do you think the teachers level of knowledge was adequate? | 4.61 |
| Q18. How was your relationship with the teachers of the workshop? | 4.89 |
| Q19. Did you enjoy the visit of the Underwater Robotics facility? | 4.33 |
| Q20. Was the additional documentation adequate and useful? | 4.33 |
| Q21. Has the activity helped you to be introduced to new technical and scientific concepts? | 4.17 |
| Q22. What is your general judgment about the activity (Good/Fair/Poor)? | |

The answers to the questions in the questionnaire can be ranked between 5 (very good) and 1 (very bad). In addition, three open-ended questions have been asked that allow the gathering of opinions on certain aspects of the activity. The answer to the second open question was yes in 88.9% of the cases. Regarding question Q22, which refers to the activity as a whole, the answer has been GOOD in 72.2% of the cases. Fortunately, thus far, no student has considered the workshop to be POOR.

Questions Q7, Q9 and Q10 obtained a score of 3 or less than 3. Q7 refers to the difficulty of building the ROV. The use of different types of techniques, materials, tools, together with the fact of combining concepts belonging to different disciplines in one device, makes the construction of an ROV difficult for students. This problem is solved by accompanying the students closely during the construction process of the ROV and explaining in detail the reason for each of the steps carried out. Questions Q9 and Q10 refer to the previous stage of preparing the work teams. Consciously, the process of preparing the work teams separates the students who belong to the same secondary school centers, with the aim that they get used to working with people they are not familiar with. This procedure is not always easy for them.

The questionnaire and the results for the teachers participating in the training and the annual final event are also shown in Table 2.

The assessment made by the teachers can also be considered as very good. The high values obtained in the answers to questions Q2 and Q12 deserve special attention. In addition, in all cases the assessment obtained from the secondary school teachers is higher than 4.25, which is very relevant to the team.

**Table 2.** Questionnaire for teachers participating on the training and annual final events.

| Questions | Score (1–5) |
|---|---|
| Q1. Enough time to build and finish the robot. | 4.62 |
| Q2. Attention received during the construction of the ROV | 5.00 |
| Q3. Coordination and organization of the final event | 4.85 |
| Q4. Evaluate facilities where the final event took place | 4.62 |
| Q5. Fulfillment of expectations with the project | 4.25 |
| Q6. Content of the subjects taught | 4.73 |
| Q7. Balance between theoretical and practical contents | 4.53 |
| Q8. Content is useful | 4.73 |
| Q9. Duration of the training | 4.38 |
| Q10. Time distribution of the training activities | 4.54 |
| Q11. Difficulty of the topics/activities | 4.73 |
| Q12. Opinion about the teachers/trainers | 4.92 |
| Q13. Group cohesion and participation | 4.85 |
| Q14. The materials used by the teachers in the training have been adequate | 4.85 |
| Q15. Facilities used during the training have been adequate | 4.85 |
| Q16. What is the general judgement about the training | 4.50 |
| Q17. Coordination and organization of the training | 4.77 |

## 5. Discussion

The number of experiences that have been carried out so far is very important. In all cases, the assessment of the experience by the students and the participating teachers is very positive. The degree of satisfaction is very high, and the activity is considered suitable for the educational purposes set out.

It should also be noted that it is not necessary to have the facilities of a research center to carry out the activity. This can be perfectly developed in any public swimming pool, and even in natural (or artificial) environments where the waters are calm. This aspect is very important for the sustainability of the project and should be specially emphasized.

The publications referring to different aspects of the activity, the research projects, the awards and distinctions received, the participation in fairs and exhibitions of different types, and the fact of having had the possibility to "export" the activity, guarantee very well the quality of the project. In addition, the activity can be considered almost unique. There is no reference to other low-cost underwater robot construction activities such as this, especially in Europe, and this is something that makes this activity very special, unique and easily exportable. That means sustainable at the end. It is possible to consider the OpenROV project [29] which makes available to the public a low cost teleoperated submarine robot, and of course, there are the extraordinary programs SeaPerch and MATE which takes place in the United States, and which, as mentioned in the introduction, has served as inspiration for the Workshops that are presented.

The project is completely linked to new ways of learning such as learning based on transversal projects, with creativity, with teamwork, focusing on different skills, and completely in accordance with the "maker" and "do-it-yourself" philosophy that is being promoted everywhere, and related to certain very interesting countercultural movements.

The result of the activity is an artifact made by the students with their own hands, which extraordinarily strengthens the link of these students with the activity and with the contents that are worked on. On the other hand, the underwater vehicle is something that it is very attractive. The submarine robot navigates underwater with a very high grace and softness, which is very surprising, in the best sense, for all the participants. What is always tried is to use low-cost materials and tools that are easy to obtain. High school technology and science teachers can easily carry out the activity. It is considered that after a few hours of training, practice and audacity, teachers can carry out the activity with complete independence. In addition, there is always the possibility of relying on the experience of the researchers for any questions the teachers in secondary schools may have about any

aspect of the activity. The researchers are always available to answer the questions coming from the colleagues.

However, carrying out this activity in an online way without direct supervision by the project researchers could suppose a significant overexertion for the trainers responsible for the student groups. The opportunities generated by the COVID-19 crisis should be taken advantage of, through creative remote learning strategies, the use of adapted pedagogical materials and videoconferencing that allows direct contact with researchers, to give full support to the remote accomplishment of the activity, although all this requires effort and represents the emergence of new challenges for young people, trainers, and families.

Finally, it is very important to value as excellent the relationship, synergy and mutual trust that exists between the secondary schools where the experience has been carried out and the members of the research groups, after all this time participating in EDUROV project. Working in joint projects between the pre-university and the university centers, is something that is necessary to promote because the extraordinary beneficial to students, researchers, teachers and all the community.

## 6. Conclusions

It can be considered that the short- or medium-term objectives of the EDUROV project are the same as those of the Education for Sustainability in the sense of developing students and schools that are able to think critically, innovate and provide solutions towards more sustainable patterns of living. It is realistic to think that it is possible to expand the target audience to which the project is directed, thus increasing sustainability. Working with young people with special educational needs (hospital classrooms, labor schools, . . . ), families, scholar groups in other countries, and establishing close relationships with similar projects (such as Seapearch or MATE programs) is the next step that must be taken to move in the right direction. In addition to promote science and technology, a new dimension of education for global citizenship must also be added to the project.

Of course, there are previous initiatives towards the internationalization of the project. We conducted two workshops in two schools for underprivileged children in South India [30,31], and there is also the Macaronesia EDUROV initiative addressed to High School Education students of Madeira and Macaronesia. Previous contacts have been established over recent years with different groups of students belonging to the Seaperch program in Puerto Rico.

On the other hand, the use of low-cost materials and easy availability of the materials used allows carrying out building underwater vehicles without major difficulties. In addition, the designs and materials used can always be modified so that the activity can be adapted to the specific needs and local conditions of each student group.

The possibility of remote teleoperation of underwater vehicles, together with the collaboration between groups of students in different locations that are in contact through these online tools, should allow the real possibility of working on the project regardless of the local conditions of these groups in many different ways. We can imagine, as an example, that a group of long-term hospitalized girls and boys from a hospital in Las Palmas de Gran Canaria could drive the underwater robots built by a group of students from a Girona secondary school at the Research Center in Underwater Robotics of the University of Girona.

This is only an example, but of course, it is possible to imagine other kind of ways of working and collaboration, among different groups of students, trainers and even researchers of the project, in the same place or remotely. The only limit is our imagination.

**Author Contributions:** Conceptualization, X.C., E.Q. and D.V.; methodology, all the authors; software, X.C., A.F., E.M., E.Q. and D.V.; validation, all the authors; formal analysis, all the authors; investigation, X.C., A.F., E.M., R.C., E.Q. and D.V.; resources, X.C., A.F., E.M., E.Q., D.V., J.L., J.J.G. and J.H.B.; data curation, X.C., E.Q. and D.V.; writing—original draft preparation, all the authors; writing—review and editing, X.C., E.Q. and D.V.; visualization, X.C., E.Q. and D.V.; supervision, X.C., R.C., E.Q. and D.V.; project administration, X.C., E.Q., D.V., J.L., J.J.G. and J.H.B.; funding acquisition,

X.C., E.Q., D.V., J.L., J.J.G. and J.H.B. All authors have read and agreed to the published version of the manuscript.

**Funding:** This research was funded by different grants from Patronat Escola Politècnica Superior de la Universitat de Girona (APAU), Consell Social de la Universitat de Girona, Ayudas ACDC Generalitat de Catalunya, Fundación Ciencia y Tecnologia FECyT (FCT-16-11269 ROVINO and FCT-17-11873 ROVSTEAM projects) and Caixabank.

**Institutional Review Board Statement:** Not applicable.

**Informed Consent Statement:** Not applicable.

**Data Availability Statement:** Data available on request.

**Conflicts of Interest:** The authors declare no conflict of interest.

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
