# Peer review of "EDUROVs: A Low Cost and Sustainable Remotely Operated Vehicles Educational Program"

_sustainability, doi:10.3390/su13158657_

Round 1

Reviewer 1 Report

The manuscript describes educational activities involving design and construction of remotely operating underwater vessels. Despite the activities are based of oversimplified tools and facilities, this is a timely and interesting work. I hope that this work will be later taken on the next level, for example, by incorporating issues concerned with operational stability, persistence to external factors, that is major consideration in the ocean. I recommend publication.

Author Response

Reviewer #1: Comments and suggestions for Authors

  1. The manuscript describes educational activities involving design and construction of remotely operating underwater vessels. Despite the activities are based of oversimplified tools and facilities, this is a timely and interesting work. I hope that this work will be later taken on the next level, for example, by incorporating issues concerned with operational stability, persistence to external factors, which is major consideration in the ocean. I recommend publication.

We would like to thank the reviewer for the comments. The issues that the reviewer are requesting are related to the most technical part of the activity and they can be found in the second paragraph of Discussion section (lines 454-457) and in the references that describes the development of the workshops ([12], [19] [20] and [22]). Of course, these issues may be considered more deeply in the future.

It should also be noted that it is not necessary to have the facilities of a research center to carry out the activity. This can be perfectly developed in any public swimming pool, and even in natural (or artificial) environments where the waters are calm. This aspect is very important for the sustainability of the project, and should be specially emphasized.

Reviewer 2 Report

Thank you very much for the opportunity to read this article. I consider the theme very important and with possible great impact on education.

I recommend the authors to adjust the research design, hypotheses and methods clearly stated. In the method section, they presented the EDUROVs Project and some info about the robot. I would recommend creating a new section: theoretical and practical background, for these issues.

In the Material and methods they might present the survey,

  • issues regarding the representativeness of the sample,
  • how did they designed the survey (Some questions are not very relevant from my point of view; which were the items analyzed, such as student satisfaction (Qx, Qy, Qz), impact on student' creativity and critical thinking(Qa, Qb, Qc....), difficulty to understand theoretical concepts explained during the workshop adequate and practical implementation, etc (Qd, Qe, Qf....)
  • hypothesis such as: The theoretical concepts and practical implementation has a great impact in stimulating creativity...et
  • tests (Cronbach, correlation, Anova, etc)

In the results section, the authors might present the test results with interpretation and in discussion to present the impact/ sustainability of the results. 

For example, the paragraph "However, carrying out this activity..." (line 425-455) might be part of the discussion. This idea can be mention in the conclusion too, but very short.

The paragraph "Of course, there are previous initiatives towards the internationalization ..." could be presented in the section Limitations and further research.

References should be updated with papers published in notoriety studies in the last 3 years.

English> line 50 "marine biodiversity to the development of developing countries, in particular small island developing States and least developed countries. "  - Development = evolution/expansion...

Once again: the article is very interesting, but the information has to be presented in a different manner.

Author Response

Reviewer #2: Comments and suggestions for Authors

  1. I recommend the authors to adjust the research design, hypotheses and methods clearly stated. In the method section, they presented the EDUROVs Project and some info about the robot. I would recommend creating a new section: theoretical and practical background, for these issues.

We would like to thanks the reviewer for the recommendations that we tried to apply to our article. We added a new subsection called theoretical and practical background (line: 101), to clarify section Introduction.

  1. In the Material and methods they might present the survey, issues regarding the representativeness of the sample, how did they designed the survey (Some questions are not very relevant from my point of view; which were the items analyzed, such as student satisfaction (Qx, Qy, Qz), impact on student' creativity and critical thinking(Qa, Qb, Qc....), difficulty to understand theoretical concepts explained during the workshop adequate and practical implementation, etc (Qd, Qe, Qf....) hypothesis such as: The theoretical concepts and practical implementation has a great impact in stimulating creativity...et tests (Cronbach, correlation, Anova, etc).

We would like to thank the reviewer for the suggestions to enhance the Material and Methods section. We have added the following paragraphs to clarify these aspects within the article, as you will see in subsequent comments. In this way, we believe that the research has gained strength in clarity and order in data collection.

“As it is shown in the Results section two surveys are presented in order to consider the perception of students and teachers participating on workshops and program events respectively. With respect the representativeness of the sample, the population considered in this study has been 2 million students, as it is the total amount of secondary school students in the Canary Islands at the time the survey was conducted.

Equation (1) describes the minimum representative sample given a targeted confidence level (which provides a score value (z)), margin of error (ε), population proportion (p) and population size (N). For a confidence value of 95% (1.96 score), a margin of error of 5% and an assumed a population proportion of 50%, the minimum sample needed to be representative would be of 384 students. In case of students, the result can be applied in any case to a higher population, as the considered population is so high that the denominator in the formula can be considered as “1”.

(1)

The survey developed for this study was shared with students, achieving a sample of 384 individuals for students. In terms of teachers, there is not a clear value of population of teachers conducting these types of activities in the classroom, but the sample value of 205 teachers taken into consideration seems enough for maximum population of 400 teachers.

The surveys were designed in order to evaluate the following aspects:

  • Students survey
    • Logistics and Facilities: Questions 1-5.
    • Underwater robotics topics: Questions 6 and 12-14.
    • Activities: Questions 7-9.
    • Teamwork: Questions 10-11.
    • Materials and Tools: Questions 15-16.
    • Teachers: Questions 17-18.
    • General assessment: Questions 19-22.
  • Teachers survey
    • Timing: Questions 1, 7, 9 and 10.
    • Organization: Questions 2-3.
    • Facilities: Questions 4 and 15.
    • Project and contents: Questions 5, 6, 8 and 11.
    • Trainers and Students: Questions 12-14.
    • General assessment: Questions 16-17.

To validate the reliability of these surveys as suitable data collecting instruments, Cronbach´s alpha method was implemented obtaining coefficients higher than 0.65 in all cases. These scores are considered as adequate according to authors such as Huh et al [27] consider that reliability value in an exploratory research should be equal or higher than 0.6. Other authors, as for instance Nunnally [28], states that, for early stages of a research, a value of 0.5 or 0.6 would be sufficient. Therefore, the surveys used as instruments in this paper count on a high reliability rate.

 [27] Huh, J.; DeLorme, D.E.; Reid, L.N. Perceived third-person effects and consumer attitudes on prevetting and banning DTC advertising. Journal of Consumer Affairs 2006, 40, 90–116. 10.1111/j.1745-6606.2006.00047.x.

[28] Nunnally, J.C. Psychometric Theory; New York: McGraw-Hill, 1967;

We want to thank the reviewer for the contributions made in order to present the information in a more appropriate way throughout the article.

  1. In the results section, the authors might present the test results with interpretation and in discussion to present the impact/ sustainability of the results.

We added this paragraph in the section Results (lines 425-433):

Questions Q7, Q9 and Q10 obtained a score of 3 or less than 3. Q7 refers to the difficulty of building the ROV. The use of different types of techniques, materials, tools, together with the fact of combining concepts belonging to different disciplines in one device, makes the construction of an ROV difficult for students. This problem is solved by accompanying the students closely during the construction process of the ROV, and explaining in detail the reason for each of the steps carried out. Questions Q9 and Q10 refer to the previous stage of preparing the work teams. Consciously, the process of preparing the work teams separates the students who belong to the same secondary school centers, with the aim that they get used to working with people they are not familiar with. This procedure is not always easy for them.

And, also related to the sustainability of the proposal, we added also this comment in section Discussions (lines 454-457). You can see also the comment addressed to reviewer 1:

It should also be noted that it is not necessary to have the facilities of a research center to carry out the activity. This can be perfectly developed in any public swimming pool, and even in natural (or artificial) environments where the waters are calm. This aspect is very important for the sustainability of the project, and should be specially emphasized.

  1. For example, the paragraph "However, carrying out this activity..." (line 425-455) might be part of the discussion. This idea can be mention in the conclusion too, but very short.

According to the suggestion of the reviewer, this paragraph has been moved to section Discussion (lines 488 – 494)

  1. The paragraph "Of course, there are previous initiatives towards the internationalization ..." could be presented in the section Limitations and further research.

There is not the section Limitations and further research in this article, for this reason we decided to leave this paragraph in section Conclusions. In fact the Conclusions section have exactly the same sense.

  1. References should be updated with papers published in notoriety studies in the last 3 years.

According to these suggestions we added two references: [6] 2018 (line 574), and [10] 2021 (line 584).

  1. English> line 50 "marine biodiversity to the development of developing countries, in particular small island developing States and least developed countries. " - Development = evolution/expansion...

We corrected this mistake, rewriting the sentence and part of the paragraph, thank you very much.

Conserve and sustainably use the oceans, seas and marine resources is the 14th Sustainable Development Goal [2]. One of the target of this Goal, specifically the tenth, is increase scientific knowledge, develop research capacity and transfer marine technology, in order to improve ocean health and to enhance the contribution of marine biodiversity to the evolution of developing countries.

  1. Once again: the article is very interesting, but the information has to be presented in a different manner.

Thank you very much for the comments and suggestions. An effort has been made to present the information in a manner consistent with what was stated by the reviewer, while at the same time considering the contributions of the rest of the reviewers.

Reviewer 3 Report

Section 1: The authors did not clearly indicate the contributions of this work. More details need to be added to the last paragraph of section 1 to explain the main contributions of this research work.

Line 207, avoid the use of words such we, I, and us

Line 211, if figure 1 has reference they should be added in the caption as well (copyright might be needed)

Line 253 “To help educational centers to achieve all these objectives”, what objectives? The above bullets are not elements of the project and not objectives.

Table 1, authors should provide ways of improvement for questions with a score less than 3.

Generally:

The paper needs to be reviewed for English and grammar.

Avoid using phrases such as “in our opinion, we believe”

The paper is lacking novel contributions as such work has been done before

The paper is lacking a scientific explanation of the design models presented in figure 1.

Author Response

Reviewer #3: Comments and suggestions for Authors

  1. Section 1: The authors did not clearly indicate the contributions of this work. More details need to be added to the last paragraph of section 1 to explain the main contributions of this research work.

We would like to thank the reviewer for the suggestions that helps a lot to improve our article, in general. It is about analyzing the perception of students and teachers to an activity that may seem complex but that becomes a challenge for both and they find it interesting and captivating.

We have added the following paragraph to clarify the contributions of the EDUROVs project (lines 207 – 220).

The EDUROVs project consists is complemented by carrying out a wide variety of activities whose central axis is to support the holding of workshops with students, and to appropriately disseminate the project in educational centers, to the scientific community, and to the public in general:

  • Teacher training sessions
  • Visits to secondary schools to help to develop the project
  • Demonstrations in public museums for the general public
  • Participation in international and national technology fairs
  • Meetings and events of secondary schools participating in the project
  • Conferences for the general public
  • Final degree projects
  • High school research works
  • Labor Stays for students in the framework of the project

  1. Line 207, avoid the use of words such we, I, and us

We revise, and we tried to avoid the use of this kind of expressions in the whole article.

  1. Line 211, if figure 1 has reference they should be added in the caption as well (copyright might be needed)

Thank you very much. We solve this for Figure 3 (line 304)

  1. Line 253 “To help educational centers to achieve all these objectives”, what objectives? The above bullets are not elements of the project and not objectives.

We have changed the wording of this sentence (lines 281 -284)

In order to help educational centers develop these improvements proposed during this 2nd phase to achieve all these objectives, a new version of the handbook was produced including all the new contents related to the integration of electronic prototyping platforms and the evolution of the project [220].

  1. Table 1, authors should provide ways of improvement for questions with a score less than 3.

We added this paragraph in the section Results (lines 425-433). You can see also the comment addressed to reviewer 2:

Questions Q7, Q9 and Q10 obtained a score of 3 or less than 3. Q7 refers to the difficulty of building the ROV. The use of different types of techniques, materials, tools, together with the fact of combining concepts belonging to different disciplines in one device, makes the construction of an ROV difficult for students. This problem is solved by accompanying the students closely during the construction process of the ROV, and explaining in detail the reason for each of the steps carried out. Questions Q9 and Q10 refer to the previous stage of preparing the work teams. Consciously, the process of preparing the work teams separates the students who belong to the same secondary school centers, with the aim that they get used to working with people they are not familiar with. This procedure is not always easy for them.

Generally:

  1. The paper needs to be reviewed for English and grammar. Avoid using phrases such as “in our opinion, we believe”

We did the revision of the whole paper in this sense, trying to avoid the use of these kind of sentences.

  1. The paper is lacking novel contributions as such work has been done before

As far as we know, it is the first underwater robotics project with these characteristics that has been carried out in times of a pandemic. You can see also the paragraph (lines 462-468) that belongs to Discussion section:

There is have no reference to other low-cost underwater robot construction activities like this, especially in Europe, and this is something that makes this activity very special, unique and easily exportable. That means sustainable at the end. It is possible to can talk about consider the OpenROV project [294] which makes available to the public a low cost teleoperated submarine robot, and of course, there are the extraordinary programs SeaPerch and MATE which takes place in the United States, and which, as mentioned in the introduction, has served as inspiration for the Workshops that are presented.

  1. The paper is lacking a scientific explanation of the design models presented in figure 1.

We added the following paragraph when we presented the Figure 1. More information about this point can be found in references that describes the development of the workshops ([12], [19] [20] and [22]).

Underwater vehicles are built with a very basic set of considerations in mind. The chassis must be able to house the 2 horizontal motors and the vertical motor, and it must be able to house in the upper part the floating elements that are required to balance the vehicle underwater. The size is that of a shoe box, approximately. This size is appropriate given the power of the DC motors used.  It is considered that this is the 1st phase of the EDUROVs project.

Round 2

Reviewer 2 Report

I think the paper  might be published!

Reviewer 3 Report

No further comments